META-RESEARCH

# Weak evidence of country- and institution-related status bias in the peer review of abstracts

**Abstract** Research suggests that scientists based at prestigious institutions receive more credit for their work than scientists based at less prestigious institutions, as do scientists working in certain countries. We examined the extent to which country- and institution-related status signals drive such differences in scientific recognition. In a preregistered survey experiment, we asked 4,147 scientists from six disciplines (astronomy, cardiology, materials science, political science, psychology and public health) to rate abstracts that varied on two factors: (i) author country (high status vs lower status in science); (ii) author institution (high status vs lower status university). We found only weak evidence of country- or institution-related status bias, and mixed regression models with discipline as random-effect parameter indicated that any plausible bias not detected by our study must be small in size.

**MATHIAS WULLUM NIELSEN\*, CHRISTINE FRIIS BAKER, EMER BRADY, MICHAEL BANG PETERSEN AND JENS PETER ANDERSEN**

## Introduction

The growth in scientific publishing (*Larsen and von Ins, 2010*) makes it increasingly difficult for researchers to keep up-to-date with the newest trends in their fields (*Medoff, 2006*; *Collins, 1998*). Already in the 1960s, sociologists of science suggested that researchers, in the midst of this information overload, would search for cues as to what literature to read (*Merton, 1968*; *Crane, 1965*). One important cue was the author's location. Academic affiliations would cast a "halo-effect" (*Crane, 1967*) on scholarly work that would amplify the recognition of researchers based at prestigious institutions at the expense of authors from institutions and nations of lower status. This halo effect ties closely to the idea of the "Matthew effect" in science, i.e. "the accumulation of differential advantages for certain segments of the population [of researchers] that are not necessarily bound up with demonstrated differences in capacity" (*Merton, 1988*).

From a social psychological perspective, country- and institution-related halo effects may arise from stereotypic beliefs about the relative performance capacity of scientists working at more or less prestigious institutions (*Jost et al., 2009*; *Greenwald and Banaji, 1995*; *Ridgeway, 2001*). According to status characteristics theory, a scientist's nationality or institutional affiliation can be considered "status signals" that, when salient, implicitly influence peer-evaluations (*Berger et al., 1977*; *Correll and Bernard, 2015*). Moreover, system justification theory asserts that members of a social hierarchy, such as the scientific community, regardless of their position in this hierarchy, will feel motivated to see existing social arrangements as fair and justifiable to preserve a sense of predictability and certainty around their own position (*Jost et al., 2004*; *Jost et al., 2002*; *Magee and Galinsky, 2008*; *Son Hing et al., 2011*). As a result, both higher and lower status groups may internalize implicit behaviours and attitudes that favour higher-status groups (*Jost, 2013*).

Several observational studies have examined the influence of country- and institution-related halo effects in peer-reviewing (*Crane, 1967*; *Link, 1998*; *Murray et al., 2018*). Most of them indicate slight advantages in favor of researchers

**\*For correspondence:** mwn@soc.ku.dk

from high status countries (such as the US or UK) and universities (such as Harvard University or Oxford University). However, a key limitation of this literature concerns the unobserved heterogeneity attributable to differences in quality. Non-experimental designs do not allow us to determine the advantages enjoyed by scientists at high-status locations independent of their capabilities as scientists, or the content and character of their work.

Here we examine the impact of academic affiliations on scientific judgment, independent of manuscript quality. Specifically, we consider how information about the geographical location and institutional affiliation of authors influence how scientific abstracts are evaluated by their peers. In a preregistered survey experiment, we asked 4147 scientists from six disciplines (astronomy, cardiology, materials science, political science, psychology and public health) to rate abstracts that vary on two factors: (i) author country (high status vs. lower status scientific nation); (ii) institutional affiliation (high status vs. lower status university; see *Table 1*). All other content of the discipline-specific abstracts was held constant.

A few pioneering studies have already attempted to discern the influence of national and institutional location on scholarly peer-assessments, independent of manuscript quality. One study (*Blank, 1991*) used a randomized field experiment to examine the effects of double blind vs. single blind peer reviewing on acceptance rates in the *American Economic Review*. The study found no evidence that a switch from single blind to double blind peer reviewing influenced the relative ratings of papers from high-ranked and lower-ranked universities. Another study (*Ross et al., 2006*) examined the effect of single blind vs. double blind peer reviewing on the assessment of 13,000 abstracts submitted to the American Heart Association's annual Scientific Sessions

between 2000 and 2004. The study found that when abstracts underwent single blind compared to double blind reviewing the relative increase in acceptance rates was higher for US authored abstracts compared to non-US authored abstracts, and for abstracts from highly prestigious US institutions compared to abstracts from non-prestigious US institutions.

A recent survey experiment also found that professors at schools of public health in the US (N: 899) rated one abstract higher on likelihood of referral to a peer, when the authors' affiliation was changed from a low-income to a high-income country (*Harris et al., 2015*). However, each participant was asked to rate four abstracts and the results for the remaining three abstracts were inconclusive. Likewise, the study found no evidence of country-related bias in the ratings of the strength of the evidence presented in the abstracts. In another randomized, blinded crossover study (N: 347), the same authors found that changing the source of an abstract from a low-income to a high-income country slightly improved English clinicians' ratings of relevance and recommendation to a peer (*Harris et al., 2017*). Finally, a controlled field experiment recently examined the "within subject" effect of peer-review model (single blind vs. double blind) on the acceptance rates of full-length submissions for a prestigious computer-science conference (*Tomkins et al., 2017*). The study allocated 974 double blind and 983 single blind reviewers to 500 papers. Two single blind and two double blind reviewers assessed each paper. The study found that single blind reviewers were more likely than double blind reviewers to accept papers from top-ranked universities compared to papers from lower-ranked universities.

Our study contributes to this pioneering work by targeting a broader range of disciplines in the social sciences, health sciences and natural sciences. This allows us to examine possible

**Table 1.** Sample distributions for the three-way factorial design across the six disciplines.
Number of observations, N, by manipulation (rows) and disciplines (columns).

| Manipulation/Discipline | Astronomy (N = 502) | Cardiology (N = 609) | Mat. science (N = 546) | Pol. science (N = 1008) | Psychology (N = 624) | Public health (N = 732) |
|---|---|---|---|---|---|---|
| Higher status (US) | N = 209 | N = 191 | N = 196 | N = 351 | N = 216 | N = 241 |
| Lower status (US) | N = 192 | N = 213 | N = 187 | N = 319 | N = 205 | N = 237 |
| Lower status (non-US) | N = 191 | N = 205 | N = 163 | N = 338 | N = 213 | N = 254 |

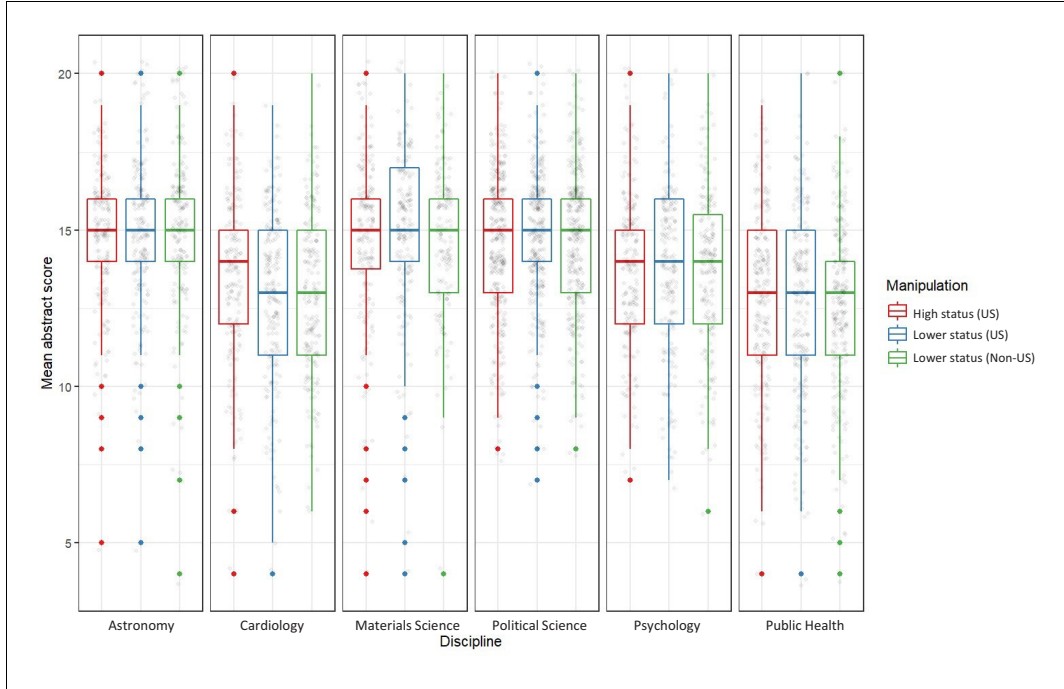

**Figure 1.** Boxplots displaying the distributions of abstract scores across the three manipulations and six disciplines. Each panel reports results for a specific discipline. The red box plots specify results for respondents exposed to an abstract authored at a high-status institution in the US. The blue box plots specify results for respondents exposed to an abstract authored at a lower-status institution in the US. The green box plots specify results for respondents exposed to an abstract authored at a lower-status institution outside the US. Whiskers show the 1.5 interquartile range. The red, blue and green dots represent outliers, and the grey dots display data points. The boxplots do not indicate any notable variations in abstract scores across manipulations.

The online version of this article includes the following figure supplement(s) for figure 1:

**Figure supplement 1.** Rating abstracts on "Originality of the presented research".
**Figure supplement 2.** Rating abstracts on "Credibility of the results".
**Figure supplement 3.** Rating abstracts on "Significance for future research".
**Figure supplement 4.** Rating abstracts on "Clarity of the abstract".
**Figure supplement 5.** The distribution of abstract scores for the full sample.

between-discipline variation in the prevalence of country- or institution-related rater bias.

Our six discipline-specific experiments show weak and inconsistent evidence of country- or institution-related status bias in abstract ratings, and mixed regression models indicate that any plausible effect must be small in size.

## Results

In accordance with our preregistered plan (see https://osf.io/4gjwa), the analysis took place in two steps. First, we used ANOVAs and logit models to conduct discipline-specific analyses of how country- and institution-related status signals influence scholarly judgments made by peer-evaluators. Second, we employed mixed regression models with disciplines as random effect parameter to estimate the direct effect

and moderation effects of the presumed association between status signals and evaluative outcomes at an aggregate, cross-disciplinary level.

We used three measures to gauge the assessments of abstracts by peer-evaluators. *Abstract score* is our main outcome variable. It consists of four items recorded on a five-point scale (1="very poor", 5 = "excellent") that ask the peer-evaluators to assess (i) the originality of the presented research, (ii) the credibility of the results, (iii) the significance for future research, and (iv) the clarity and comprehensiveness of the abstract. We computed a composite measure that specifies each participant's total-item score for these four items (Cronbach's $\alpha$ = 0.778). *Open full-text* is a dichotomous outcome measure that asks whether the peer-evaluator would choose to open the full text and continue reading, if s/he came across the abstract online.

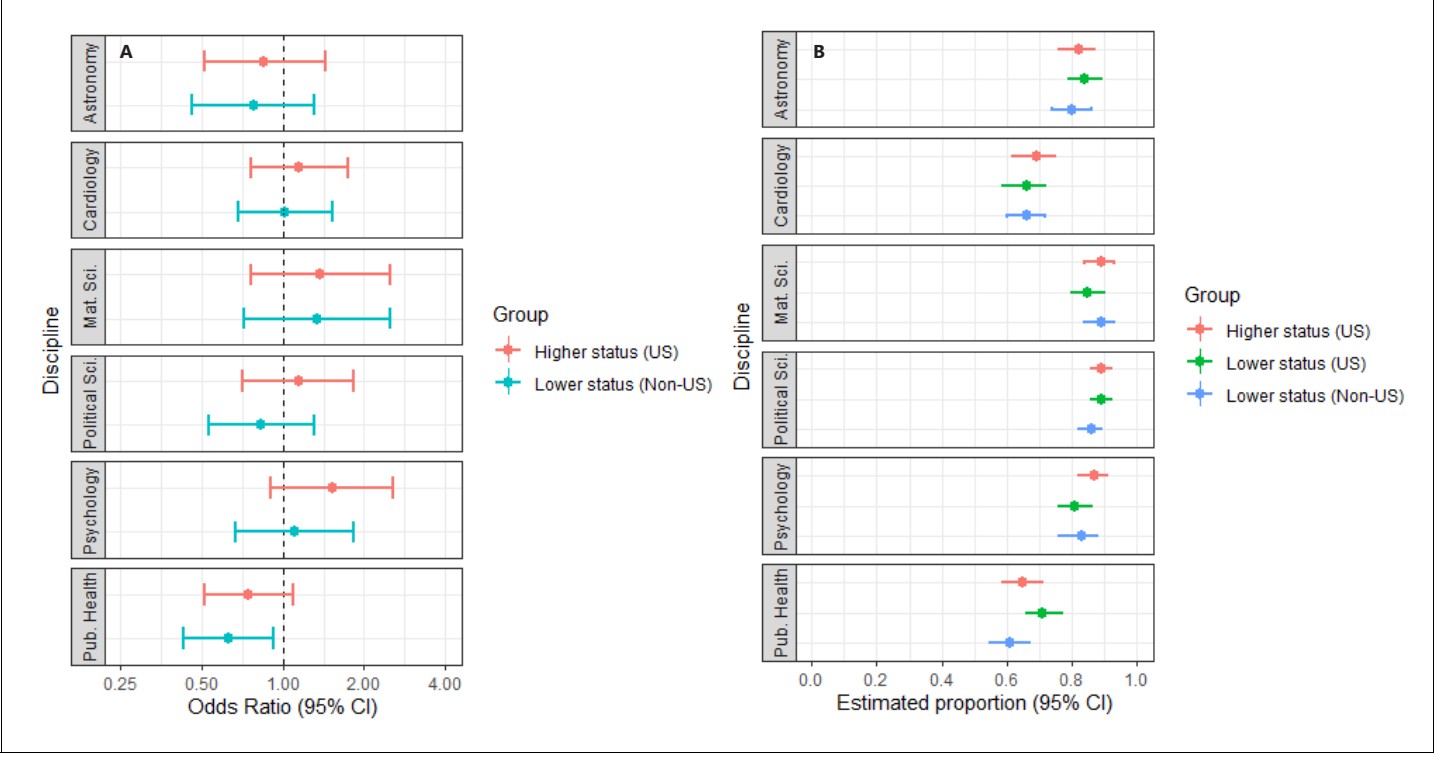

**Figure 2.** Plots of odds ratios and estimated proportions derived from the discipline-specific logit models with "*Open full text*" as outcome. Panel A displays the odds ratios for respondents exposed to manipulation 1 (high-status university, US) or manipulation 3 (lower-status university, non-US). Manipulation 2 (lower-status university, US) is the reference group. Panel B plots the adjusted means for manipulation 1, manipulation 2 and manipulation 3. Error bars represent 95% CIs. As shown in Panel A, peer-evaluators in public health were slightly less likely to show interest in opening the full text when the author affiliation was changed from a lower-status university in the US to a lower-status university elsewhere. The results for the remaining eleven comparisons are inclusive. For model specifications, see *Supplementary file 2*, Tables S13–S18.

*Include in conference* is a dichotomous outcome measure that asks whether the peer-evaluator would choose to include the abstract for an oral presentation, if s/he was reviewing it as a committee member of a selective international scientific conference (the full questionnaire is included in the *Supplementary file 1*).

As specified in the registered plan, we used G*Power to calculate the required sample size per discipline (N = 429) for detecting a Cohen's $f$ = 0.15 (corresponding to a Cohen's $d$ = 0.30) or larger with α = 0.05 and a power of 0.80 in the cross-group comparisons with abstract score as outcome. *Table 1* presents the final sample distributions for the three-way factorial design across the six disciplines.

To test for statistical equivalence in the discipline-specific comparisons of our main outcome (abstract score) across manipulations, we performed the two-one-sided tests procedure [TOST] for differences between independent means (*Lakens, 2017a*). We used the pre-registered 'minimum detectable effect size' of

Cohen's $D$=±0.30 for the discipline-specific equivalence tests.

The boxplots in *Figure 1* display the abstracts ratings for respondents assigned to each manipulation across the six disciplines. The discipline-specific ANOVAs with the manipulations as the between-subject factor did not indicate any country- or institution-related status bias in abstract ratings (astronomy: F = 0.71, p=0.491, N = 592; cardiology: F = 1.50, p=0.225, N = 609; materials science: F = 0.73, p=0.482, N = 546; political science: F = 0.53, p=0.587, N = 1008; psychology: F = 0.19, p=0.827, N = 624; public health: F = 0.34, p=0.715, N = 732). The $\eta^2$ coefficients were 0.002 in astronomy, 0.005 in cardiology, 0.003 in materials science, and 0.001 in political science, psychology and public health.

A TOST procedure with an α level of 0.05 indicated that the observed effect sizes were within the equivalence bound of $d = -0.3$ and $d = 0.3$ for 11 of the 12 between-subject comparisons at the disciplinary level (see *Supplementary file 2*, Tables S1–S12). In raw

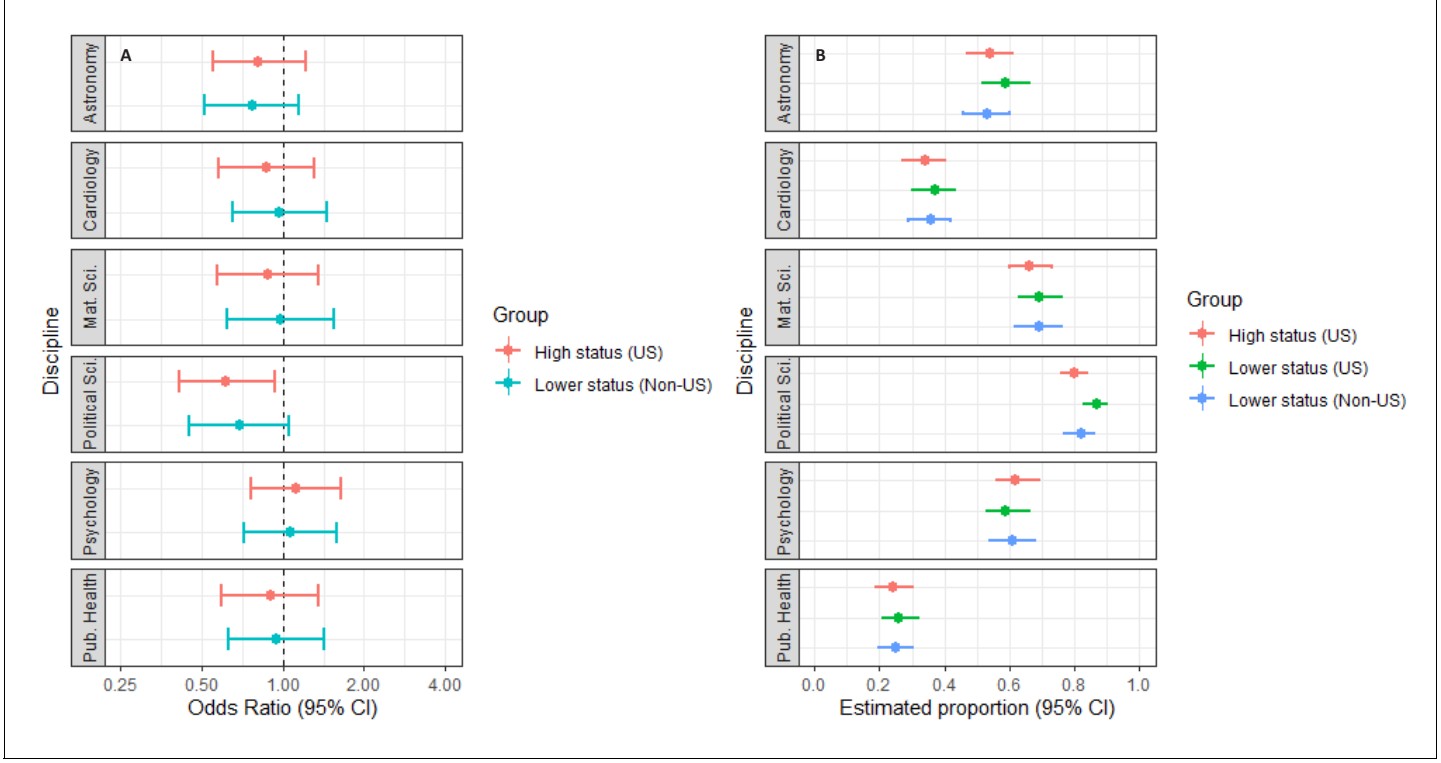

**Figure 3.** Plots of odds ratios and estimated proportions derived from the discipline-specific logit models with "*Include in conference*" as outcome. Panel **A** displays the odds ratios for respondents exposed to manipulation 1 (high-status university, US) or manipulation 3 (lower-status university, non-US). Manipulation 2 (lower-status university, US) is the reference group. Panel **B** plots the adjusted means for manipulation 1, manipulation 2 and manipulation 3. Error bars represent 95% CIs. As shown in Panel A, peer-evaluators in political science were slightly less likely to show interest in opening the full text when the author affiliation was changed from a lower-status university in the US to a high-status university in the US. The results for the remaining eleven comparisons are inclusive. For model specifications, see *Supplementary file 2*, Tables S19–S24.

scores, this equivalence bound corresponds to a span from −0.8 to 0.8 abstract rating points on a scale from 4 to 20. In cardiology, the TOST test failed to reject the null-hypothesis of non-equivalence in the evaluative ratings of subjects exposed to abstracts from higher-status US universities and lower-status US universities.

A closer inspection suggests that these findings are robust across the four individual items that make up our composite abstract-score. None of these items show notable variations in abstract ratings across manipulations (*Figure 1—figure supplement 1–4*).

*Figures 2* and *3* report the outcomes of the discipline-specific logit models with *Open full-text* and *Include in conference* as outcomes. The uncertainty of the estimates is notably larger in these models than for the one-way ANOVAs reported above, indicating wider variability in the peer-evaluators' dichotomous assessments.

As displayed in *Figure 2*, peer-evaluators in Public Health were between 7.5% and 56.5% less likely to show interest in opening the full-text, when the author affiliation was changed

from a lower-status university in the US to a lower-status university elsewhere (Odds ratio:. 634, CI: 0.435–0.925). The odds ratios for the remaining 11 comparisons across the six disciplines all had 95% confidence intervals spanning one. Moreover, in five of the 12 between-subject comparisons, the direction of the observed effects was inconsistent with the a-priori expectation that abstracts from higher-status US universities would be favoured over abstracts from lower-status US universities, and that abstracts from lower-status US universities would be favoured over abstracts from lower-status universities elsewhere.

As displayed in *Figure 3*, peer-evaluators in political science were between 7.0% and 59.4% less likely to consider an abstract relevant for a selective, international conference program, when the abstract's author was affiliated with a higher-status US university compared to a lower-status US university (Odds ratio:. 613, CI:. 406-.930). This result goes against a-priori expectations concerning the influence of status signals on evaluative outcomes. The remaining five

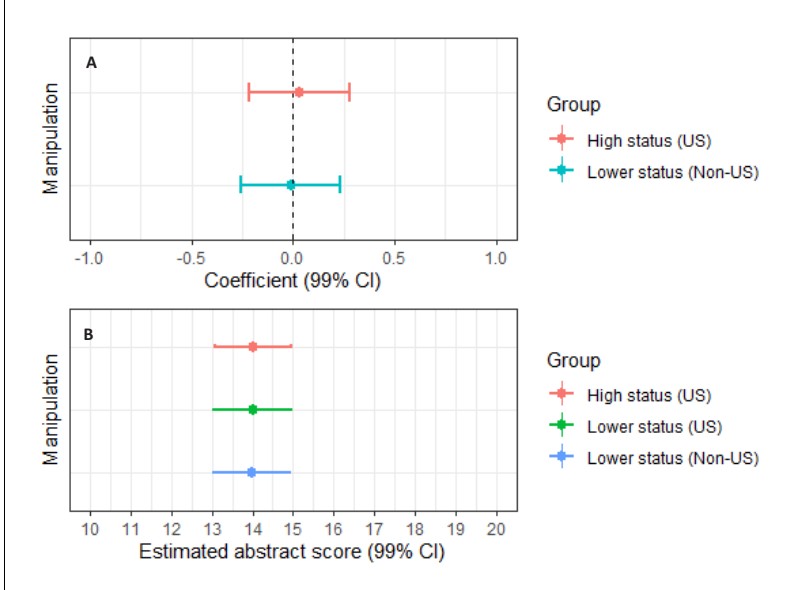

**Figure 4.** Plots of fixed coefficients and adjusted means derived from the mixed-linear regression model with "*Abstract score*" as outcome. Panel A plots the fixed coefficients for manipulation 1 (high-status university, US) and manipulation 3 (lower-status university, non-US). Manipulation 2 (lower-status university, US) is the reference group. Panel B plots the adjusted means for manipulation 1, manipulation 2 and manipulation 3. Error bars represent 99% CIs. The figure shows that status cues in the form of institutional affiliation or national affiliation have no tangible effects on the respondents' assessments of abstracts. For model specifications, see *Supplementary file 2*, Table S25.

disciplines had 95% confidence intervals spanning one, and in six of the 12 comparisons, the direction of the observed effects was inconsistent with a-priori expectations. As indicated in panel b, we observe notable variation in the participants' average responses to the *Include in conference* item across disciplines.

*Figure 4* plots the fixed coefficients (panel a) and adjusted means (panel b) from the mixed linear regression with *Abstract rating* as outcome. In accordance with the pre-registered analysis plan, we report the mixed regression models with 99% confidence intervals. The fixed coefficients for the two between-group comparisons tested in this model are close to zero, and the 99% confidence intervals suggest that any plausible difference would fall within the bounds of −0.26 to 0.28 points on the 16-point abstract-rating scale. These confidence bounds can be converted into standardized effects of Cohen's $d = -0.10$ to 0.11.

*Figure 5* displays odds ratios and 99% confidence intervals for the mixed logit regressions with *Open full-text* (panel a, upper display) and *Include in conference* (panel a, lower display) as outcomes. The odds ratios for the experimental manipulations used as predictors in these

models range from 0.86 to 1.05 and have 99% confidence intervals spanning the line of no difference. The 99% confidence intervals (panel a) indicate that any plausible effect would fall within the bounds of odds ratio = 0.68 and 1.35, which corresponds to a standardized confidence bound of Cohen's $d = -0.21$ to 0.17. The wide confidence bounds for the estimated proportion for *Include in conference* (panel b) reflect the large variations in average assessments of abstracts across disciplines.

Robustness checks based on mixed linear and logit models were carried out to examine the effects of the experimental manipulations on the three outcome measures, while restricting the samples to (i) participants that responded correctly to a manipulation-check item, and (ii) participants that saw their own research as being 'extremely close', 'very close' or 'somewhat close' to the subject addressed in the abstract. All of these models yielded qualitatively similar results, with small residual effects and confidence intervals spanning 0 in the linear regressions and one in the logistic regressions (see *Supplementary file 2*, Tables S28–S33). Moreover, a pre-registered interaction analysis was conducted to examine whether the influence of country- and institution-related status signals was moderated by any of the following characteristics of the peer evaluators: (i) their descriptive beliefs in the objectivity and fairness of peer-evaluation; (ii) their structural location in the science system (in terms of institutional affiliation and scientific rank); (iii) their research accomplishments; (iv) their self-perceived scientific status. All of these two-way interactions had 99% CI intervals spanning 0 in the linear regressions and one in the logistic regressions, indicating no discernible two-way interactions (see *Supplementary file 2*, Tables S34–S39).

## Discussion

Contrary to the idea of halo effects, our study shows weak and inconsistent evidence of country- or institution-related status bias in abstract ratings. In the discipline-specific analyses, we observed statistically significant differences in two of 36 pairwise comparisons (i.e. 5.6%) and, of these, one was inconsistent with our a-priori expectation of a bias in favour of high-status sources. Moreover, the estimated confidence bounds derived from the multilevel regressions were either very small or small according to Cohen's classification of effect sizes (*Cohen, 2013*). These results align with the

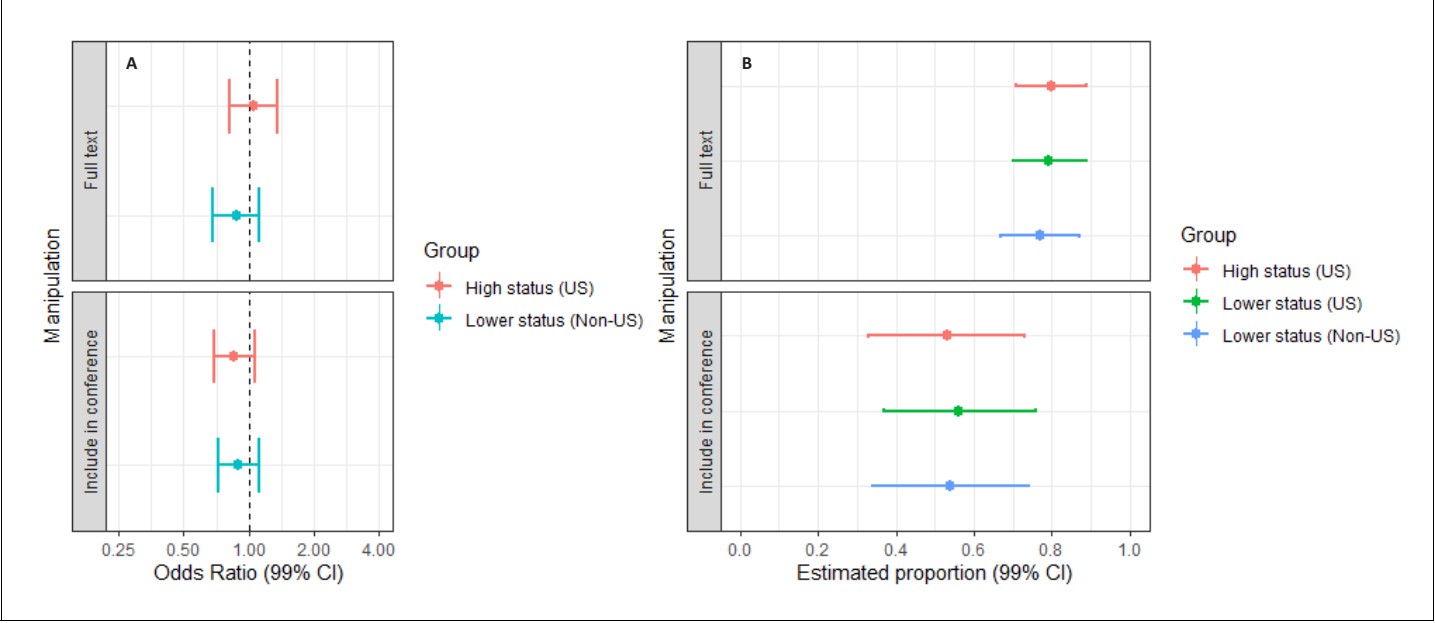

**Figure 5.** Plots of odds ratios and adjusted means derived from the mixed logit regressions with "*Open full text*" (upper part) and "*Include in conference*" (lower part) as outcomes. Panel **A** displays the odds ratios for respondents exposed to manipulation 1 (high-status university, US) or manipulation 3 (lower-status university, non-US). Manipulation 2 (lower-status university, US) is the reference group. Panel **B** plots the adjusted means for manipulation 1, manipulation 2 and manipulation 3. Error bars represent 99% CIs. As shown in Panel A, the results for both regression models are inconclusive, and the effect sizes are small. For model specifications, see *Supplementary file 2*, Tables S26–27.

outcomes of three out of five existing experiments, which also show small and inconsistent effects of country- and institution-related status bias in peer-evaluation (*Blank, 1991*; *Harris et al., 2015*; *Harris et al., 2017*). However, it should be noted that even small effects may produce large population-level variations if they accumulate over the course of scientific careers. Moreover, one could argue that small effects would be the expected outcome for a lightweight manipulation like ours in an online survey context, where the participants are asked to make decisions without real-world implications. Indeed, it is possible that the effects of country- and institution level status signals would be larger in real-world evaluation processes, where reviewers have more at stake.

Certain scope conditions also delimit the situations to which our conclusions may apply. First, our findings leave open the possibility that peer-evaluators discard research from less reputable institutions and science nations without even reading their abstracts. In other words, our study solely tests whether scientists, after being asked to carefully read an assigned abstract, on average will rate contributions from prestigious locations more favourably.

Second, even *when* peer-evaluators identify and read papers from "lower-status" sources, they may still omit to cite them, and instead frame their contributions in the context of more high-status sources. In the future, researchers might add to this work by examining if these more subtle processes contribute to shape the reception and uptake of scientific knowledge. Third, our conclusion of little or no bias in abstract review does not necessarily imply that biases are absent in other types of peer assessment, such as peer reviewing of journal and grant submissions, where evaluations usually follow formalized criteria and focus on full-text manuscripts and proposals. Likewise, our study does not capture how country- and institution-related status signals may influence the decisions of journal editors. Indeed, journal editors play an important role in establishing scientific merits and reputations and future experimental work should examine how halo effects may shape editorial decisions. Fourth, although we cover a broad span of research areas, it is possible that our experiment would have produced different results for other disciplines. Fifth, it should be noted that our two dichotomous outcome items (open full text and include in conference) refer to two rather different evaluative situations that may be difficult to compare. For instance, a researcher may wish to read a paper (based on its abstract) while finding it inappropriate for a

conference presentation and vice versa. Moreover, the competitive aspect of selecting an abstract for a conference makes this evaluative situation quite different from the decision to open a full text.

A key limitation is that our experiment was conducted in non-probability samples with low response rates, which raises questions about selection effects. One could speculate that the scientists that are most susceptible to status bias would be less likely to participate in a study conducted by researchers from two public universities in Denmark. Moreover, we decided to include internationally outstanding universities in our high-status category (California Institute of Technology, Columbia University, Harvard University, Yale University, Johns Hopkins University, Massachusetts Institute of Technology, Princeton University, Stanford University, and University of California, Berkeley). By so doing, we aimed to ensure a relatively strong status signal in the abstract's byline. There is a risk that this focus on outstanding institutions may have led some survey participants to discern the purpose of the experiment and censor their responses to appear unbiased. However, all participants were blinded to the study objectives (the invitation email is included in *Supplementary file 1*), and to reduce the salience of the manipulation, we asked each participant to review only one abstract. Moreover, in the minds of the reviewers, many other factors than the scholarly affiliations might have been manipulated in the abstracts, including the gender of the authors (*Knobloch-Westerwick et al., 2013*; *Forscher et al., 2019*), the style of writing, the source of the empirical data presented in the abstract, the clarity of the statistical reporting, the reporting of positive vs. negative and mixed results (*Mahoney, 1977*), the reporting of intuitive vs. counter-intuitive findings (*Mahoney, 1977*; *Hergovich et al., 2010*), and the reporting of journal information and visibility metrics (e.g. citation indicators or Altmetric scores) (*Teplitskiy et al., 2020*). A final limitation concerns the varying assessments of abstract quality across disciplines. This cross-disciplinary variation was particularly salient for the *Include in conference* item (*Figure 3*, panel b), which may indicate differences in the relative difficulty of getting an oral presentation at conferences in the six disciplines. In the mixed-regression models based on data from all six disciplines, we attempt to account for this variation by including discipline as random-effect parameter. In summary, this paper presents a large-scale, cross-disciplinary examination of how country- and institution-related status signals influence the reception of scientific research. In a controlled experiment that spanned six disciplines, we found no systematic evidence of status bias against abstracts from lower status universities and countries. Future research should add to this work by examining if other processes related to the social, material and intellectual organization of science contribute to producing and reproducing country- and institution-related status hierarchies.

## Methods
Our study complies with all ethical regulations. Aarhus University's Institutional Review Board approved the study (case no. 2019-616-000014). We obtained informed consent from all participants. The sampling and analysis plan was pre-registered at the Open Science Framework on September 18, 2019. We have followed all of the steps presented in the registered plan, with two minor deviations. First, we did not preregister the equivalence tests reported for the discipline-specific analyses presented in *Figure 1*. Second, in the results section, we report the outcomes of the mixed regression models with abstract score as outcome based on linear models instead of the tobit models included in the registered plan. Tables S40–S44 in *Supplementary file 2* report the outcomes of the mixed-tobit models, which are nearly identical to the results from the linear models reported in Tables S25, S28, S31, S36, S37 in *Supplementary file 2*.

### Participants
The target population consisted of research-active academics with at least three articles published between 2015 and 2018 in Clarivate's Web of Science (WoS). To allow for the retrieval of contact information (specifically email addresses), we limited our focus to corresponding authors with a majority of publications falling into one of the following six disciplinary categories: astronomy, cardiology, materials science, political science, psychology and public health. These disciplines were chosen to represent the top-level domains; natural science, technical science, health science, and social science. We did not include the arts and humanities as the majority of fields in this domain have very different traditions of publishing and interpret scholarly quality in less comparable terms. While other fields could have been chosen as representative

of those domains, practical aspects of access to field experts and coverage in Web of Science were deciding for the final delineation. We used the WoS subject categories and the Centre for Science and Technology Studies' (CWTS) meso-level cluster classification system to identify eligible authors within each of these disciplines. For all fields, except for materials science, the WoS subject categories provided a useful field delineation. In materials science, the WoS subject categories were too broad. Hence, we used the article-based meso-level classification of CWTS to identify those papers most closely related to the topic of our abstract. In our sampling strategy, we made sure that no participants were asked to review abstracts written by (fictive) authors affiliated with their own research institutions.

We used G*Power to calculate the required sample size for detecting a Cohen's $f$ = 0.15 (corresponding to a Cohen's $d$ = 0.30) or larger with $\alpha$ = 0.05 and a power of 0.80 in the discipline-specific analyses with abstract rating as outcome. With these assumptions, a comparison of three groups would require at least 429 respondents per discipline. Response rates for e-mail-based academic surveys are known to be low (*Myers et al., 2020*). Based on the outcomes of a pilot study targeting neuroscientists, we expected a response rate around 5% and distributed the survey to approximately 72,000 researchers globally (i.e. approximately 12,000 researchers per field) (for specifications, see *Supplementary file 2*, Table S45). All data were collected in October and November 2019. Due to low response rates in materials science and cardiology, we expanded the recruitment samples by an additional ~12,000 scientists in each of these disciplines. In total, our recruitment sample consisted 95,317 scientists. Each scientist was invited to participate in the survey by email, and we used the web-based Qualtrics software for data collection. We sent out five reminders and closed the survey two weeks after the final reminder. Eight percent (N = 7,401) of the invited participants opened the email survey link, and about six percent (N = 5,413) completed the questionnaire (for specifications on discipline-specific completion rates, see *Supplementary file 2*, Table S45). For ethical reasons, our analysis solely relies on data from participants that reached the final page of the survey, where we debriefed about the study's experimental manipulations. The actual response rate is difficult to estimate. Some scientists may have refrained from participating in the study because they did not see themselves as belonging to one of the targeted disciplines. Others may not have responded because they were on sabbatical, parental leave or sick leave. Moreover, approximately 16 percent (15,247) of the targeted email addresses were inactive or bounced for other reasons. A crude estimate of the response rate would thus be 5,413/(95,317–15,247)=0.07, or seven percent. The gender composition of the six respondent samples largely resembles that of the targeted WoS populations (*Supplementary file 2*, Table S46). However, the average publication age (i.e. years since first publication in WoS) is slightly higher in the respondent samples compared to the targeted WoS populations, which may be due to the study's restricted focus on corresponding authors (the age distributions are largely similar across the recruitment and respondent samples). The distribution of respondents (per discipline) across countries in WoS, the recruitment sample, and the respondent sample is reported in *Supplementary file 2*, Table S47.

### Pretesting and pilot testing

Prior to launching, the survey was pretested with eight researchers in sociology, information science, political science, psychology, physics, biomedicine, clinical medicine, and public health. In the pretesting, we used verbal probing techniques and "think alouds" to identify questions that the participants found vague and unclear. Moreover, we elicited how the participants arrived at answers to the questions and whether the questions were easy or hard to answer, and why.

In addition, we pilot-tested the survey in a sample of 6,000 Neuroscientists to (i) estimate the expected response rate per discipline, (ii) check the feasibility of a priming instrument that we chose not to include in the final survey, (iii) detect potential errors in the online version of the questionnaire before launching, and (iv) verify the internal consistency of two of the composite measures used in the survey (i.e. abstract score and meritocracy beliefs).

### Procedures

In each of the six online-surveys (one per discipline), we randomly assigned participants to one of the three manipulations (*Table 1*). All participants were blinded to the study objectives.

## Manipulations

We manipulated information about the abstract's institutional source (high status vs. lower status US research institution) and country source (lower status US research institution vs. lower-status research institution in a select group of European, Asian, Middle Eastern, African and South American countries). The following criteria guided our selection of universities for the manipulation of institutional affiliation: Candidates for the "high status" category all ranked high in the US National Research Council's field-specific rankings of doctorate programs, and consistently ranked high (Top 20) in five subfield-specific university rankings (US News' graduate school ranking, Shanghai ranking, Times Higher Education, Leiden Ranking and QS World University Ranking).

The universities assigned to the "lower status" category were selected based on the PP-top 10% citation indicator used in the Leiden University Ranking. For non-US universities, we limited our focus to less esteemed science nations in Southern Europe, Eastern Europe, Latin America, Asia, Africa and the Middle East. Institutions assigned to the "lower-status" category were matched (approximately) across countries with respect to PP-top 10% rank in the field-specific Leiden University ranking. We decided to restrict our focus to universities in the Leiden ranking to ensure that all fictive affiliations listed under the abstracts represented research active institutions in the biomedical and health sciences, the physical sciences and engineering, or the social sciences. By matching lower-status universities on their PP-top 10% rank, we ensured that the lower-status universities selected for each discipline had a comparable level of visibility in the scientific literature. Since a given university's rank on the PP-top 10% indicator may not necessarily align with its general reputation, we also ensured that none of the lower-status universities were within the top-100 in the general Shanghai, Times Higher Education and QS World University rankings.

Given this approach, the specific institutions that fall into the "high status" and "lower status" categories vary by discipline. We carried out manual checks to ensure that all of the selected universities had active research environments within the relevant research disciplines. The universities assigned to each category (high-status [US], lower-status [US], lower-status [non-US]), and the average abstract scores per university, per discipline, are listed in *Supplementary file 2*. Table S48, S49.

The abstracts were created or adapted for this study and are not published in their current form. The abstracts used in astronomy, materials science, political science and psychology were provided by relevant researchers in the respective disciplines and have been slightly edited for the purposes of this study. The abstracts used in cardiology and public health represent rewritten versions of published abstracts with numerous alterations to mask any resemblance with the published work (the six abstracts are available in *Supplementary file 1*). Author names were selected by searching university websites for each country and identifying researchers in disciplines unrelated to this study.

## Measures

Variable specifications are reported in *Supplementary file 2*, Table S50. The outcome variables used in this analysis are specified above. We used dichotomous variables to estimate the effect of the manipulations on the outcomes in all regression models. We used the following measures to compute the moderation variables included in the two-way interaction analyses. Our measure of the respondents' descriptive beliefs in the objectivity and fairness of peer-evaluation in their own research field (i.e. *meritocratic beliefs*) was adapted from ref (*Anderson et al., 2010*). A sample item from this measure reads: "In my research field, scientists evaluate research primarily on its merit, i.e. according to accepted standards of the field". We adapted the sample item from ref (*Anderson et al., 2010*). The two other items were developed for this study. Ratings were based on a five-point scale ranging from (1) 'Strongly agree' to (5) 'Strongly disagree'. Based on these items, we computed a composite measure that specifies each participant's total-item score for these three items (i.e. meritocratic beliefs) (Cronbach's $\alpha$ = 0.765). We used two pieces of information to measure the participants' structural location in the science system (i.e. *structural location*): (i) information about scientific rank collected through the survey, and (ii) information about scientific institution obtained from Web of Science. Our measure of structural location is dichotomous. Associate professors, full professors, chairs and deans at top ranked international research institutions are scored as 1; all other participants are scored as 0. Here, we define top-ranked research institutions as institutions that have consistently ranked among

the top 100 universities with the highest proportion of top 10% most cited papers within the past 10 years, according to the Leiden Ranking. We used article metadata from WOS to construct an author-specific performance profile for each respondent (i.e. *research accomplishments*). Specifically, we assigned researchers that were among the top-10% most cited in their fields (based on cumulative citation impact) to the "high status" group. All other participants were assigned to the "lower status" group. Our measure of *self-perceived status* was adapted from the MacArthur Scale of Subjective Social Status (*Adler and Stewart, 2007*). We asked the respondents to locate themselves on a ladder with ten rungs representing the status hierarchy in their research area. Respondents that positioned themselves at the top of the ladder were scored as 9, and respondents positioning themselves at the bottom were scored as 0.

### Manipulation and robustness checks

As a manipulation check presented at the end of the survey, we asked the participants to answer a question about the author's institutional affiliation/country affiliation in the abstract that they just read. The question varied depending on manipulation and discipline. For robustness checks, we included an item to measure the perceived distance between the participant's own research area and the topic addressed in the abstract. Responses were based on a five-point scale ranging from (1) 'Extremely close' to (5) 'Not close at all'. In the statistical analysis, these response options were recoded into dichotomous categories ('Not close at all', 'Not too close'=0, 'Somewhat close', 'Very close', 'Extremely close'=1).

### Data exclusion criteria

In accordance with our registered plan, respondents that demonstrated response bias (10 items of the same responses, e.g. all ones or fives) were removed from the analysis. Moreover, we removed all respondents that completed the survey in less than 2.5 min.

### Statistical analysis

We used one-way ANOVAs and logit models to perform the discipline-specific, between-group comparisons. We estimated mixed linear regressions and tobit models (reported in *Supplementary file 2*) with disciplines as random effect parameter to measure the relationship between the experimental manipulations and *abstract rating*. The tobit models were specified with a left-censoring at four and a right-censoring at 20. *Figure 1—figure supplement 5* displays the data distribution for the outcome measure *abstract rating*. The data distribution for this measure was assumed to be normal.

We estimated multilevel logistic regressions with disciplines as random effect parameter to examine the relationship between the manipulations and the outcome variables *Open full-text* and *Include in conference*. Consistent with our pre-registered analysis plan, we used 95% confidence intervals to make inferences based on the discipline specific ANOVAs and logistic regressions. To minimize Type I errors arising from multiple testing, we reported the results of all multilevel regression models with 99% confidence intervals.

We created two specific samples for the moderation analyses. The first of these samples only include respondents that had been exposed to abstracts from a high status US university or a lower status US university. The second sample was restricted to respondents that had been exposed to abstracts from a lower status US university or a lower status university outside the US. The Variance Inflation Factors for the predictors included in the moderation analyses (i.e. the manipulation variables, Meritocracy beliefs, Structural Location, Research Accomplishments and Self-perceived status) were all below two.

We conducted the statistical analyses in STATA 16 and R version 4.0.0. For the multilevel linear, tobit and logit regressions, we used the "mixed", "metobit" and "melogit" routines in STATA. Examinations of between-group equivalence were performed with the R package 'TOSTER' (*Lakens, 2017b*). Standardized effects were calculated using the R package 'esc' (*Lüdecke, 2018*).

## Acknowledgements

The Centre for Science and Technology Studies (CWTS) at Leiden University generously provided bibliometric indices and article metadata. We thank Emil Bargmann Madsen, Jesper Wiborg Schneider and Friedolin Merhout for very useful comments on the manuscript.

**Mathias Wullum Nielsen** is in the Department of Sociology, University of Copenhagen, Copenhagen, Denmark

mwn@soc.ku.dk

https://orcid.org/0000-0001-8759-7150

**Christine Friis Baker** is in the Danish Centre for Studies in Research and Research Policy, Department

of Political Science, Aarhus University, Aarhus, Denmark

🆔 https://orcid.org/0000-0002-3370-021X

**Emer Brady** is in the Danish Centre for Studies in Research and Research Policy, Department of Political Science, Aarhus University, Aarhus, Denmark

🆔 https://orcid.org/0000-0002-6065-8096

**Michael Bang Petersen** is in the Department of Political Science, Aarhus University, Aarhus, Denmark

🆔 https://orcid.org/0000-0002-6782-5635

**Jens Peter Andersen** is in the Danish Centre for Studies in Research and Research Policy, Department of Political Science, Aarhus University, Aarhus, Denmark

🆔 https://orcid.org/0000-0003-2444-6210

*Author contributions:* Mathias Wullum Nielsen, Conceptualization, Resources, Data curation, Formal analysis, Supervision, Funding acquisition, Validation, Investigation, Visualization, Methodology, Writing - original draft, Project administration, Writing - review and editing; Christine Friis Baker, Emer Brady, Formal analysis, Investigation, Visualization, Methodology, Writing - review and editing; Michael Bang Petersen, Formal analysis, Validation, Investigation, Methodology, Writing - review and editing; Jens Peter Andersen, Conceptualization, Data curation, Formal analysis, Funding acquisition, Validation, Investigation, Visualization, Methodology, Project administration, Writing - review and editing

*Competing interests:* The authors declare that no competing interests exist.

*Ethics:* Human subjects: Aarhus University's Institutional Review Board approved the study. We obtained informed consent from all participants (case no. 2019-616-000014).

**Funding**

| Funder | Grant reference number | Author |
|---|---|---|
| Carlsbergfondet | CF19-0566 | Mathias Wullum Nielsen |
| Aarhus Universitets Forskningsfond | AUFF-F-2018-7-5 | Christine Friis Baker Emer Brady Jens Peter Andersen |

The funders had no role in study design, data collection and interpretation, or the decision to submit the work for publication.

**Decision letter and Author response**
Decision letter https://doi.org/10.7554/eLife.64561.sa1

Author response https://doi.org/10.7554/eLife.64561.sa2

## Additional files
### Supplementary files
• Supplementary file 1. Qualtrics survey (astronomy); the six abstracts used in the survey experiment; email invitation.
• Supplementary file 2. Tables S1-S50.
• Transparent reporting form

### Data availability
All data and code needed to evaluate the conclusions are available here: https://osf.io/x4rj8/.

The following dataset was generated:

| Author(s) | Year | Dataset URL | Database and Identifier |
|---|---|---|---|
| Nielsen MW | 2020 | https://osf.io/x4rj8/ | Open Science Framework, x4rj8 |

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
