## [Decision Letter]

Thank you for submitting your article "Weak evidence of institutional and country-related bias in experimental study of 4,000 scientists' research assessments" to *eLife* for consideration as a Feature Article. Your article has been reviewed by three peer reviewers, and the following individual involved in review of your submission has agreed to reveal their identity: Bjorn Hammarfelt (Reviewer #1).

In view of the reviewers comments, you are invited to prepare a revised submission that addresses their comments, please see below.

Summary:

This is a carefully designed study and a well-written paper about bias in peer review based on authors' institutional and country location. The survey design is novel and the results seem robust. The main results show weak evidence for the claim that researchers are biased towards high-status countries and institutions. The presentation is logical and the paper is well-structured, and the main arguments are clearly articulated. However, there are also a number of points that need to be addressed to make the article suitable for publication.

Essential revisions:

1) Why choose the very top universities – like Harvard and MIT – rather than top universities that are less likely to be identified as the classic examples of “the best”. Is there a risk that participants of the survey would realize that this was part of the experiment, and thus acting accordingly in terms of “social desirability”. Perhaps a short discussion of these choices, maybe in the limitations section, would be useful.

2) Similarly, I wonder why there are no "Elite (non-US)" institutions (e.g. Oxford, ETH, etc.). Would not the inclusion of such universities make sense given that you include Elite (US) elite, and Non-elite (US) as well as Non-elite (non-US)? As it stands the selection might come off as a bit too focused on an American context, while the participants in the survey are from the entire world.

3) Please list somewhere in the article or supporting material the countries and institutions included in the survey that fall into each category and field.

4) The authors divide institutions into three categories. If I understood well, high US institutions are those in the top 20 of several discipline based rankings and in high positions in the US NRC doctorate programs ranking. Low status are those with low share of highly cited papers according to Leiden Ranking, and the same for non-US non-Elite from certain regions. While rankings may be a plausible way to go, I am not sure to what extent using the PP-top 10% score will help identifying low status institutions. In this case, the authors are looking for institutions with lower reputation (which not necessarily aligns with low citation performance), furthermore, the PP-top indicator is non-size independent, and therefore will work against larger institutions. Also, the Leiden Ranking ranks around 1000 institutions worldwide which fit certain publication threshold, probably including institutions that do not make it to the Leiden Ranking would give a better idea of low status.

5) I wonder to what extent the difference between the three categories is sufficiently detailed as to be able to show any differences. My concern is that you divide in so few groups that in the end differences may be hidden within those three large groups of institutions.

6) Effect sizes

The authors find that effect of manipulations on abstract quality ratings is likely not larger than the band -0.8 to 0.8 (16 pt scale) (Results). That does seem small, although in Cohen's d terms, d=-0.3 to 0.3 isn't all that small. Overall, we would expect effects of a light-weight manipulation in an online survey context to be tiny, but it's really hard to generalize such interventions to the field, where status effects may be large and subtle. So I'd like a little more language emphasizing low-ish (but informative) external validity here, and that the prior on light weight manipulation is very small effects.

7) You control for the reliability on the peers' judgment (how close is the abstract to their area of expertise). Why did you not do something similar with the institution as a means to control on the appropriateness of the category assignment of each institution? Could this be affecting also the results?

8) I understand that the six fields/disciplines were selected in order to cover a range of different disciplines, but a few paragraphs about why these particular fields were chosen would be useful.

9) Please discuss further the difference between choosing to read the full text and to include it in a conference. These are rather different evaluative situations: one may want to read a paper that one would not accept for a conference and vice versa. For example, a researcher might deem a paper as interesting, yet have doubts about the quality, or judge it as “worthy” of presentation while not being interested in reading it. You discuss that the disciplinary differences were higher when choosing to include in a conference or not (Discussion), and I would add that the competitive aspect of selecting for a conference makes this choice rather different than choosing to read a paper or not. This could be elaborated upon in the text and perhaps be something to study closer in future research.

10) In the framing of the paper the authors refer to biases on peer review but also discuss the Matthew Effect. While there is a relation between the halo effect that the authors are exploring and the Matthew Effect, the latter refers to the reception of publications (in terms of citations and prestige) while the former refers to biases in peer review. I think this difference is not sufficiently clear in the text, especially when reading the Appendix Text. I'd say that the fact that an halo effect may not be present in peer review would not necessarily reflect on a Matthew Effect or vice versa.

11) Situating this work in the existing literature.

I think the part of the literature review section that tries to distinguish this contribution from others isn't entirely fair. Here are the distinguishing points, followed by my comments on them:

a) Separating out location status vs. institution status

Comment: Why would this separating out really matter? Surely status of country is highly correlated with status of institution, and, presumably, the mechanism at work are inferences of quality from status. It seems fine to me to separate them out, but would be good to have an argument for why this is valuable to do

b) Previous RCTs in the peer review context may have been hampered by treatment leakage to the control group.

Comment: This surely happens ( https://arxiv.org/abs/1709.01609 ). But it just means the reported effects in those RCTs are lower bounds, as those papers I believe readily admit. So I don't think this has been a major methodological problem in dire need of fixing.

So I don't think what this experiment is doing is super novel and fixes glaring oversights in the literature. BUT, that isn't a bad thing, in my view. Simply adding observations, particularly those that are well executed, adequately powered, and cross-disciplinary, is important enough. It's an important topic, so we should have more than just one or two RCTs in it. Crucially, we know there are file-drawer effects out there, likely in this space too, so this study is useful for making sure the published record is unbiased.

---

## [Author Response]

Essential revisions:1) Why choose the very top universities – like Harvard and MIT – rather than top universities that are less likely to be identified as the classic examples of “the best”. Is there a risk that participants of the survey would realize that this was part of the experiment, and thus acting accordingly in terms of “social desirability”. Perhaps a short discussion of these choices, maybe in the limitations section, would be useful.

This is an excellent point. We now reflect on why we made this choice, and the possible limitations of this choice, in the Discussion section.

2) Similarly, I wonder why there are no "Elite (non-US)" institutions (e.g. Oxford, ETH, etc). Would not the inclusion of such universities make sense given that you include Elite (U.S) elite, and Non-elite (US) as well as Non-elite (non-US)? As it stands the selection might come off as a bit too focused on an American context, while the participants in the survey are from the entire world.

This is a fair point. We decided to restrict the category of elite universities to the U.S. for the following reasons. (1) Our design allowed us to compare within-country variations in university status, and hereby tease apart the effect of country status and institution status. If we had included Oxford, Cambridge, ETH and University of Amsterdam, this would have required a more comprehensive set of lower-ranked universities in the UK, Switzerland and Netherlands as well, which in turn would have made our setup more complex. In other words, we wanted to keep the experiment simple. (2) 15 of the top 20 universities in the 2019 THE ranking, and 10 of the top 20 institutions in the 2019 US News ranking were located the U.S. Given this ratio, we found it justifiable to restrict our measure of institutional halo effects to elite universities in the U.S.

3) Please list somewhere in the article or supporting material the countries and institutions included in the survey that fall into each category and field.

Thank you for this request. We have included information on countries and institutions in Supplementary file 2.

4) The authors divide institutions into three categories. If I understood well, high US institutions are those in the top 20 of several discipline based rankings and in high positions in the US NRC doctorate programs ranking. Low status are those with low share of highly cited papers according to Leiden Ranking, and the same for non-US non-Elite from certain regions. While rankings may be a plausible way to go, I am not sure to what extent using the PP-top 10% score will help identifying low status institutions. In this case, the authors are looking for institutions with lower reputation (which not necessarily aligns with low citation performance), furthermore, the PP-top indicator is non-size independent, and therefore will work against larger institutions. Also, the Leiden Ranking ranks around 1000 institutions worldwide which fit certain publication threshold, probably including institutions that do not make it to the Leiden Ranking would give a better idea of low status.

Thank you for raising this point. There is certainly a status differential between institutions included in the Leiden Ranking and those not included. However, there is also a concern that using institutions that are not internationally recognized, e.g. by being present on such a ranking, could create other sentiments, such as distrust. Our argument is that we measure a status differential because the institutions are “on the same scale”, since they can be found on these rankings, while if we had included those outside the rankings, we might end up measuring on two different scales. In this case, we decided to rely on the PP-top 10% indicator because it measures a university’s *share* of publications among the 10% most cited as opposed to the *number* of publications among the 10% most cited (i.e. P-top 10%). In the revised Materials and methods section, we reflect in more detail on these considerations.

5) I wonder to what extent the difference between the three categories is sufficiently detailed as to be able to show any differences. My concern is that you divide in so few groups that in the end differences may be hidden within those three large groups of institutions.

Thank you for raising this important point. As an attempt to address this issue, we have computed the distribution of the average abstract ratings across universities, per discipline. As shown in Supplementary table 49 in Supplementary file 2, the within-group variations are quite modest. We chose to operate with the three categories to ensure reasonable statistical power in the analysis. As shown in Supplementary table 49 in Supplementary file 2 the Ns for individual universities are too small for any meaningful statistical comparison.

6) Effect sizesThe authors find that effect of manipulations on abstract quality ratings is likely not larger than the band -0.8 to 0.8 (16 pt scale) (Results). That does seem small, although in Cohen's d terms, d=-0.3 to 0.3 isn't all that small. Overall, we would expect effects of a light-weight manipulation in an online survey context to be tiny, but it's really hard to generalize such interventions to the field, where status effects may be large and subtle. So I'd like a little more language emphasizing low-ish (but informative) external validity here, and that the prior on light weight manipulation is very small effects.

Thank you for pointing this out. We now reflect on this issue in the first paragraph of the Discussion section.

7) You control for the reliability on the peers' judgment (how close is the abstract to their area of expertise). Why did you not do something similar with the institution as a means to control on the appropriateness of the category assignment of each institution? Could this be affecting also the results?

This is an interesting point. However, we are not entirely sure how such a control measure would be operationalized in practice? As we state in the Materials and methods section, “We carried out manual checks to ensure that all of the selected universities had active research environments within the relevant research disciplines.” Moreover, our selection of high-status institutions were all ranked highly in the U.S. National Research Council’s discipline-specific rankings of doctorate programs.

8) I understand that the six fields/disciplines were selected in order to cover a range of different disciplines, but a few paragraphs about why these particular fields were chosen would be useful.

This is a great point. In the updated Materials and methods section, we outline our reasons for targeting these particular disciplines.

9) Please discuss further the difference between choosing to read the full text and to include it in a conference. These are rather different evaluative situations: one may want to read a paper that one would not accept for a conference and vice versa. For example, a researcher might deem a paper as interesting, yet have doubts about the quality, or judge it as “worthy” of presentation while not being interested in reading it. You discuss that the disciplinary differences were higher when choosing to include in a conference or not (Discussion), and I would add that the competitive aspect of selecting for a conference makes this choice rather different than choosing to read a paper or not. This could be elaborated upon in the text and perhaps be something to study closer in future research.

We completely agree. We designed the two dichotomous outcome items to capture additional types of evaluative situations. While the two dichotomous outcomes may not be comparable, we find both of them suitable for the purposes of our experiment. We have revised our discussion of these outcomes in accordance with your suggestions (please see the paragraph on scope conditions).

10) In the framing of the paper the authors refer to biases on peer review but also discuss the Matthew Effect. While there is a relation between the halo effect that the authors are exploring and the Matthew Effect, the latter refers to the reception of publications (in terms of citations and prestige) while the former refers to biases in peer review. I think this difference is not sufficiently clear in the text, especially when reading the Appendix Text. I'd say that the fact that an halo effect may not be present in peer review would not necessarily reflect on a Matthew Effect or vice versa.

This is an excellent point. In the revised Introduction, we have limited our focus to studies that relate directly to the question of halo effects.

11) Situating this work in the existing literature.I think the part of the literature review section that tries to distinguish this contribution from others isn't entirely fair. Here are the distinguishing points, followed by my comments on them:a) Separating out location status vs. institution statusComment: Why would this separating out really matter? Surely status of country is highly correlated with status of institution, and, presumably, the mechanism at work are inferences of quality from status. It seems fine to me to separate them out, but would be good to have an argument for why this is valuable to dob) Previous RCTs in the peer review context may have been hampered by treatment leakage to the control group.Comment: This surely happens ( https://arxiv.org/abs/1709.01609 ). But it just means the reported effects in those RCTs are lower bounds, as those papers I believe readily admit. So I don't think this has been a major methodological problem in dire need of fixing.So I don't think what this experiment is doing is super novel and fixes glaring oversights in the literature. BUT, that isn't a bad thing, in my view. Simply adding observations, particularly those that are well executed, adequately powered, and cross-disciplinary, is important enough. It's an important topic, so we should have more than just one or two RCTs in it. Crucially, we know there are file-drawer effects out there, likely in this space too, so this study is useful for making sure the published record is unbiased.

Thank you for raising these important points. We have revised the last part of the Introduction in accordance with your suggestions.